# Factors Associated with the Perception of Risk and Knowledge of Contracting the SARS-Cov-2 among Adults in Bangladesh: Analysis of Online Surveys

**DOI:** 10.3390/ijerph17145252

**Published:** 2020-07-21

**Authors:** Tanvir Abir, Nazmul Ahsan Kalimullah, Uchechukwu Levi Osuagwu, Dewan Muhammad Nur -A. Yazdani, Abdullah Al Mamun, Taha Husain, Palash Basak, P. Yukthamarani Permarupan, Kingsley E. Agho

**Affiliations:** 1College of Business Administration—CBA, International University of Business, Agriculture and Technology—IUBAT University, Dhaka 1230, Bangladesh; dewanm@iubat.edu; 2Vice-chancellor, Begum Rokeya University, Rangpur-5404, Bangladesh; janipop1995@gmail.com; 3Diabetes, Obesity, and Translational Research Unit (DOMTRU), School of Medicine, Western Sydney University, Sydney 2560, Australia; l.osuagwu@westernsydney.edu.au; 4Faculty of Business and Management, UCSI University, Kuala Lumpur 56000, Malaysia; abdullaham@ucsiuniversity.edu.my; 5Department of Gender and Development Studies, Begum Rokeya University, Rangpur-5404, Bangladesh; ahmthusain@gmail.com; 6School of Environment and Life Sciences (Environmental Science and Management), University of Newcastle, Callaghan 2308, Australia; palash.Basak@gmail.com; 7Faculty of Entrepreneurship and Business, Universiti Malaysia Kelantan, Kota Bharu 16100, Malaysia; yuktha@umk.edu.my; 8School of Health Sciences, Western Sydney University, Sydney 2000, Australia; K.Agho@westernsydney.edu.au

**Keywords:** COVID-19, knowledge, perception of risk, pandemic outbreak, disease control, cross-sectional study

## Abstract

This study investigated the perception and awareness of risk among adult participants in Bangladesh about Coronavirus Disease 2019 (COVID-19). During the lockdown era in Bangladesh at two different time points, from 26−31 March 2020 (early lockdown) and 11−16 May 2020 (late lockdown), two self-administered online surveys were conducted on 1005 respondents (322 and 683 participants, respectively) via social media. To examine risk perception and knowledge-related factors towards COVID-19, univariate and multiple linear regression models were employed. Scores of mean knowledge (8.4 vs. 8.1, *p* = 0.022) and perception of risk (11.2 vs. 10.6, *p* < 0.001) differed significantly between early and late lockdown. There was a significant decrease in perceived risk scores for contracting SARS-Cov-2 [β = −0.85, 95%CI: −1.31, −0.39], while knowledge about SARS-Cov-2 decreased insignificantly [β = −0.22, 95%CI: −0.46, 0.03] in late lockdown compared with early lockdown period. Self-quarantine was a common factor linked to increased perceived risks and knowledge of SARS-Cov-2 during the lockdown period. Any effort to increase public awareness and comprehension of SARS-Cov-2 in Bangladesh will then offer preference to males, who did not practice self-quarantine and are less worried about the propagation of this kind of virus.

## 1. Introduction

The increasingly busy human civilization has been interrupted by a deadly pandemic which, no matter how distinctive, threatens every nation in the world [1]. SARS-CoV-2 is a beta coronavirus genetic more closely linked to the SARS-CoV-1 (79% sequence identity) than to the MERS-CoV (51,8% identity) [2]. This virus was first detected in the city of Wuhan, a Chinese province of Hubei, in December 2019 when few patients were documenting a very different form of the acute pneumonia-like disease in that area [3]. The signs included dry cough, dyspnea, fever and lung trouble [4]. The infection quickly became a source of concern since no accepted cure or current drug was available to combat the virus [5]. SARS-Cov-2 propagated extremely rapidly, infecting millions in just a few days [6].

Whereas most of the affected persons can recover at their own, severe illness is most likely to develop amongst people of advanced age and those with co-existing conditions, including hypertension, diabetes cardiovascular disease, chronic lung disease, and obesity. The overall mortality rate for COVID-19 is likely in the range of 0.5% to 3.5% [7,8]. The most common symptoms of COVID-19 are fever and cough, and later in the disease, patients are more likely to have the difficulty breathing and develop pneumonia. At diagnosis, 80% of cases are asymptomatic or will have mild disease, 15% are severe, and 5% are considered critical [7,8]. There are currently no approved drugs, immune therapies or vaccines against SARS-CoV-2. Another real concern has been the frequency at which the infection spreads in a quite relatively short period [9], with the potential to overwhelm the healthcare system, including the likelihood of those who need emergency health attention succumbing to death as a result of overwhelming health services [10]. To delay the spread of infection and dispersing pressures on hospitals, public health authorities in most of the country adopted vigorous public health measures, such as surveillance, exhaustive contact tracing, social distancing, travel restrictions, and educating the public on hand hygiene, and postponing non-essential operations and services [11]. As previous research reveals, interpersonal distancing, personal hygiene and people’s collective consciousness inhibited SARS-Cov-2 from spreading [12,13].

The virus has now infected more than 200 countries and nearly all of them are shut down to prevent the virus from transmitting [14]. Starting from China, this virus hit European, American, African and Asian countries. Spreading continues to be an issue in Asia, where the disease began, and the spread will probably continue, even though it now seems to have reached its height in China. In South Asia, it is now quite prevalent. However, this worldwide pandemic surge has not exempted Bangladesh, a nation of approximately 170 million inhabitants, where the largest population in the world lies. The Bangladesh Government took stringent measures, including community shutdown, social isolation, and self-hygiene, to restrict the transmission of the virus [5,15,16]. Since Bangladesh reported its first case of SARS-Cov-2 infection in early March 2020, there has been a rising number of cases and deaths from the virus throughout the community [6]. 

The rapid spread of the virus has put enormous pressure on many countries’ local health care systems and, if expected to proceed, this pandemic will have profound economic consequences, together with other potential dangers [17]. Many countries, which are affected by the SARS-Cov-2 virus, are already under strict, partial, or full lockdown procedures [4]. For instance, the Chinese provinces that were affected were locked down from any kind of communication for more than three months [9]. Italy, Canada, Spain, the United States, and other countries that embarked on total lockdown were avoiding any national resurgence. Wide-scale domestic and foreign religious events have been cancelled for fear of SARS-Cov-2 outbreak [18]. Such actions have an enormous socio-economic impact on the country [15] and the shutdown, has upstretched fears of economic repercussions [19]. Because of this pandemic, everything about human life, including exports and imports of goods, business, infrastructural development, agriculture, and education seem to have stopped [18].

Economic crises may have severe effects in parts of the world, such as Bangladesh. It is predicted that more than USD 3 billion would be needed to be spent if this infectious disease causes a large outbreak in Bangladesh and, therefore, approximately 800,000 jobs in Bangladesh could be terminated [20]. This may contribute to a big economic tragedy in a country such as Bangladesh, which is still attributed to everyday wages, as seen in other heavily affected regions of the world. 

Since the sheer illness of the whole country is sufficient to destroy the health care system, this current study is to examine changes of individual perception of risk for contracting SARS-Cov-2, and the awareness level in Bangladesh during the early and late lockdowns implemented by the government of Bangladesh. The findings of this study will provide an understanding of people’s knowledge level, perception of risk and awareness which can be used to implement emergency policies to counter the spread of SARS-Cov-2.

## 2. Methodology

From 26 to 31 March 2020, the first cross-sectional survey entitled “early lockdown” was performed, referring to the week of the lockdown period in Bangladesh and the second cross-sectional survey entitled “late lockdown” was carried out from 11 to 16 May 2020. Even though a national community-based sampling survey throughout that time was not conceivable, the data were collected electronically using a Google Form. A standardized synchronized questionnaire was uploaded on social networking sites, such as Facebook and WhatsApp, which are widely used by investigators and local people throughout the country. Emails with the survey link were sent in the second step via contact lists of the researchers to broaden the scope of the survey. Participants in the survey received no incentives.

### 2.1. Sample Size

The first survey (early lockdown) assumed a 50% proportion with 90% confidence. Because the main objective of this research was on SARS-Cov-2 and there are no previous studies from Bangladesh that examined factors associated with this, an online sample size calculator [21] was used and we took a sample size of approximately 300, including a 10% non-response rate. The second survey (late lockdown) assumed a proportion of 31% (very worried about SARS-Cov-2) reported in the first study (early lockdown) with 90% confidence [21]. The calculation of the total sample size for the second survey was 710, including a 10% non-response rate. 

### 2.2. Consent and Ethical Consideration

The participants responding to a “yes” or “no” question obtained voluntary on-line consent to express their willingness to attend the study via Google forms. This study was approved by the Ethics Committee (Approval Number: BRUR/DWRTI/a.n.003) of the Dr Wazed Research and Training Institute, Begum Rokeya University, Rangpur. Rangpur-5404, Bangladesh. 

### 2.3. Questionnaire

Table 1 presents the questionnaire used in this study. The questionnaire was divided into three sections, including demographics, knowledge, and perception. The demographic variables included age, gender, marital status, education, employment, and religion. There were 12 items on the questionnaire that assessed the respondent’s knowledge of COVID- 19, most of which required a “yes” or “no” response. Each question used a binary scale. The scores for each item ranged from 0 (No) to 1 (Yes). The knowledge score ranged from 0–12 points. These items have been validated elsewhere to have an acceptable internal consistency [22]. The survey tool for the COVID-19 knowledge questionnaire was developed based on the guidelines from the World Health Organization [5,23] for clinical and community management of COVID-19.

We asked the respondents about risk perception towards COVID-19 (P1−P4). Each question used a Likert scale with five levels. The scores for each item ranged from 1 (lowest) to 5 (highest). The risk perception score ranged from 5 to 20 points. The Cronbach’s alpha coefficients of the perception items were 0.74 and demonstrated that the internal consistency of perception items was satisfactory. Respondents were also asked, “How they felt about the quarantine” (P6−P11). Each question used a Likert scale with five levels. The scores for each item ranged from 1 (lowest) to 5 (highest) and the Cronbach’s alpha coefficient of the “How they felt about the quarantine items” was 0.70, indicating acceptable internal consistency.

### 2.4. Independent Variables

The explanatory (independent) variable included basic characteristics and explanatory factors including gender, age in categories, level of education, marital, employment, and religious status. The question of worrying about quarantine score ranged from 6 to 30 points. The worried about quarantine score was divided into three categories. The bottom 33.3% of the score was arbitrarily referred to as “low quarantine practice”, the next 33.3% as “moderate quarantine practice”, and the top 33.3% as “high quarantine practice”. Furthermore, “high quarantine practice”, which was derived by combining the moderate quarantine practice (33.3%) with the high quarantine practice (33.3%) and low quarantine practice, was “low quarantine practice scores” 33.3%.

### 2.5. Statistical Analysis

Data analysis was performed using Stata version 14.1 (Stata Corp. College Station United States of America). Categorical variables were presented as frequency and percentage. This study used a t-test to compare the differences between means for early and late lockdowns for knowledge and risk perception items. In the univariate linear regression analysis, all confounding variables with a *p*-value < 0.20 were retained and used to build a multivariable linear regression model and to determine factors associated with the knowledge and perception score towards COVID-19. Additionally, we performed a similar stage modelling to that employed by Dibley et al. [24], and a two-staged modelling technique was employed in the multivariable modelling. In the first stage, the demography factors were entered into the baseline multivariable model. A manual process of backward elimination was performed, and variables with *p* < 0.05 were retained in the first model (Model 1). In the second and final stage of modelling, perceived risk of COVID-19 factors was added into significant variables in Model 1, and variables with *p*-values < 0.05 were retained in the final model. For all regression analyses performed, we checked the homogeneity of variance and multicollinearity using Variance Inflation Factors (VIF). 

## 3. Results

### 3.1. Descriptive Statistics

The descriptive statistics of the explanatory and dependent variables are shown in Table 2. This summary of responses was obtained from those who participated in the survey during the early lockdown (26–31 March 2020) and late lockdown (11–16 May 2020) periods. Total responses were a combination of both. Most of the respondents (53.2%, *n* = 532) were 18-28 years old with equal representation of males and females. Most respondents (58.2%, *n*=585) were married, and almost all (83.1%, *n* = 835) completed tertiary education or its equivalent. Of the respondents, 88.8% (*n* = 892) were Muslims, about two-thirds of them (65.5%, *n* = 658) voluntarily quarantined themselves during the study period while about a quarter of them (19.2%, *n*=193) did not quarantine. Regarding their concern for the spread of the SARS-Cov-2 virus, the majority (68.7%, *n* = 690) stated that they were very worried. 

### 3.2. Prevalence of Perceived Risk and Knowledge Towards COVID-19

Figure 1a,b show the mean and 95% Confidence Intervals of perceives risk and knowledge towards COVID-19, respectively. Data of early and late lockdown periods are presented here, correspondingly. Figure 1a indicates statistical differences between early and late lockdowns (*p* < 0.001), with early lockdown reporting the highest mean values. Additionally, as indicated in Figure 1b, knowledge towards COVID-19 for early lockdown significantly reported the highest mean value compared with late lockdown (*p* = 0.022). The horizontal values in Figure 1a,b are the minimums and maximums of perceived risk and knowledge scores. 

### 3.3. Factors Associated with the Perceived Risk of the SARS-Cov-2 Infection

The unadjusted and adjusted coefficients for factors associated with the perceived risk of contracting SARS-Cov-2 are presented in Table 3. Compared with the early lockdown period, the results indicated that perceived risk scores for contracting COVID-19 in late lockdown period reduced significantly (adjusted coefficients (β) −0.85, 95% CI:−1.31, −0.39). Other factors associated with perceived risk scores for contracting COVID-19 are females, practised high quarantine, very worried about COVID-19, and quarantined at the request of public health order during the lockdown period. Age stratification was significant in the univariate analysis and the final model, we removed religion and replaced it with age stratification, and the result showed that age stratification was not statistically significant (Wald χ2 = 0.46, *p* = 0.7137) and similarly, when gender was replaced with age stratification, age stratification was not significant (Wald χ2 = 0.49, *p* = 0.6908). The factors associated with perceived risk scores for contracting COVID-19 in early lockdown and late lockdown period are presented in Table A1 and Table A2.

### 3.4. Factors Associated with Adequate Knowledge of the SARS-Cov-2 Infection

Table 4 showed the unadjusted and adjusted coefficients with 95% confidence intervals (CIs) of the knowledge level of COVID-19. After the adjustment of potential confounding factors, knowledge about COVID-19 has decreased but it was not statistically significant [β = −0.22, 95%CI: −0.46, 0.03] in late lockdown period compared to early lockdown period. Additionally, comparatively less knowledge of COVID-19 was pertinent among those who performed low quarantine and those who had less education (completed secondary or primary education only). Increased knowledge of COVID-19 was pertinent among the participants who practised high quarantine, held a bachelor and above degree, and the non-Muslim participants. Age stratification and employment status were significant in the univariate analysis and our final model, we removed religion and replaced it with age stratification, and age stratification was not statistically significant (Wald χ2 = 1.44, *p* = 0.2293) and when education was replaced with age stratification, age stratification was not significant (Wald χ2 = 2.54, *p* = 0.055), but when education was replaced by employment status, employment status was associated with increased knowledge of COVID-19 [β = 0.26, 95%CI: 0.03, 0.49, *p* = 0.027 for those employed]. Factors associated with the knowledge level of COVID-19 for each lockdown periods are reported in Table A3 and Table A4.

## 4. Discussion

This current study reported a higher mean of perception of risk and low knowledge of contracting the SARS-Cov-2 among adults in Bangladesh. The study also revealed factors associated with the perception of risk and knowledge of contracting the SARS-Cov-2 in Bangladesh and found that females and those with a bachelor’s degree reported decreased perceived risk and knowledge of contracting SARS-Cov-2 than males, and Master’s/higher degree holders, respectively, practised high quarantine, were very worried, and quarantined at the request of public health order during COVID-19, and reported higher perceptive risk of contracting COVID-19, while non-Muslims (Christian/Hindu) practised high quarantine and quarantined at the request of public health order during COVID-19, and reported increased knowledge scores of contracting the infection.

The higher mean score of risk perceptions stated in this analysis could be because the Bangladesh government has taken exceptional measures to track the rapid spread of the current global COVID-19 disease outbreak [25]. When the number of individuals infected and the fatalities from this epidemic escalate, residents will stick to preventive measures because they are influenced by their knowledge, perceptions and practices towards this disease outbreak [26]. In this study, we analyzed the opinions of Bangladeshi people about vulnerability and awareness towards COVID-19 during the drastic rise period of the disease outbreak. Researchers identified that many were extremely concerned about the transmission of the infectious disease in this predominantly well-educated young Muslim population and more than one-third considered themselves to be at low risk of contracting the infection. Such a high perception of low risk, coupled with the fairly average COVID-19 knowledge scores, is extremely important because clear knowledge predicts a positive attitude and appropriate attitude against COVID-19 [22]. 

In this study, males who were worried about contracting SARS-Cov-2 were more likely to perceive themselves as being at high risk of contracting the infection, as well as those who did not quarantine themselves or only did so at the request of the public health officers. These findings were similar to those reported in the studies conducted in India, China, and Jordan. Adults with a higher level of knowledge about COVID-19 and who were in quarantine were more concerned about the infection and became frustrated as they did not know how long the impact of the pandemic would last [27]. Moreover, in India, it was found that a higher level of knowledge on COVID-19 was associated with the high-risk perception of contracting the infection during the consistent lockdown period [28]. In Jordan [14], it was found that, with adequate knowledge, people can perceive the importance of lockdown and the risk of contracting the infection caused by SARS-Cov-2.

Experience from previous similar virus attacks (SARS-CoV-1) in China highlighted the fact that, during such a crisis, people’s knowledge, attitudes, and perceptions about the situation affects their response to the crisis. To effectively manage a health emergency, citizens need to be conscious of the problem, to be alert, and acknowledge their responsibilities to preserve their steadiness, because circumstances culminating in fear in the public can escalate the situation into misery [22]. A similar survey conducted to test the knowledge, attitudes, and perceptions of people in the Hubei province, China, about the COVID-19 outbreak found that higher knowledge, attitude and perception scores among residents was related to the ages and socioeconomic statuses of the respondents [22]. It was surprising to find an average score of knowledge against COVID-19 among Bangladesh residents, considering that this epidemiological survey was performed at the very early stage of the pandemic in Bangladesh. We believe this to be partially attributed to the survey being skewed by people with a bachelor’s degree or higher, the largest percentage of respondents being 86%. The magnitude of this pandemic and the unprecedented media attention of this public health disaster will have an important effect on people’s awareness about this epidemic. Television channels, Bangladesh health ministry official websites, and all corporation websites had details about this infectious disease during this time. Adults with higher levels of education are more likely to seek information which enhances a sense of personal control through mastering content and acquiring stronger skills [29].

Similar to previous findings [22,30,31,32] which suggested that men and young adults are more inclined to engage in risk-taking behaviours, the present study found a significant association between male gender and perceived high-risk of COVID-19 among respondents after adjusting for other cofounders. Adults who were employed at the time of this study were 0.6 times more likely to show adequate knowledge scores compared to those who were unemployed, but this association was significant only when it interacted with other demographic variables in the model. The slightly higher chances of sufficient information among citizens who did not quarantine themselves, relative to those who did so willingly, could be due to the less severe situation of the COVID-19 outbreak in Bangladesh and the prevalence of younger adults in this sample, resulting in respondents feeling that they had a lower probability of contamination with the SARS-Cov-2 virus.

It is worth noting that, in this analysis, higher COVID-19 awareness scores are strongly correlated with not becoming a practising Muslim. It is understood that the negative mentality shown by certain religious manipulators is one of the toughest obstacles in attempts to tackle the dissemination of COVID-19 awareness. While the government has called for the public to keep social distances to stop the gathering of crowds (physical distance), certain so-called religious leaders might also be preparing to host meetings involving hundreds. Resistance from religious communities to physical isolating appeals has been observed across several predominantly Muslim countries, such as Indonesia, and the trend exacerbates local government attempts to negotiate with COVID-19 propagation. Research in Turkey [33] echoed the significance of religious figures throughout this disease outbreak in positively motivating populations. Although some practitioners preferred to seek counsel from their municipal officials, others adopted their religious leader’s instructions when it came to debatable questions, such as COVID-19, suggesting that religious leaders have strong influence on the respondent’s attitude towards COVID-19 mitigation practices during the pandemic.

The finding of this study indicates the value of strengthening public health knowledge for Bangladeshi citizens towards COVID-19. This, in effect, would change behaviours and activities towards COVID-19. Research findings of the demographic variables correlated with knowledge towards SARS-Cov-2 are broadly compatible with previous research on SARS-CoV-1 in 2003 [16,22], further indicating that the intervention in health education towards COVID-19 in Bangladesh would become more successful if it had been primarily structured for mass people and those with low educational thresholds.

Since SARS-Cov-2 is a new type of coronavirus, and no pharmacologic therapies at this time are available, increased public awareness and caution seem to be the best approaches to preventing community spread. The travel bans and lockdowns placed in many countries, including Bangladesh, may have worked, but they also raised the level of panic among residents. This was evident in this study, where approximately 31% of the respondents were very worried, and others were somewhat worried about the situation. In this situation, Lai and others showed that educating the public is a very helpful and effective resource [34]. For countries with fragile health care systems who have dense populations, such as the sub-Saharan African countries, lack of awareness about the virus and corrupt policies can combine to create a disaster that is impossible to contain [35]. In the case of COVID-19, issues with the current response, lack of transparency, travel restriction delay, quarantine delay, public misinformation, and emergency announcement delay contributed to the outbreak.

## 5. Conclusions

The findings of this study show that many of the respondents in Bangladesh were very worried about the spread of COVID-19 coupled with their significant inadequacies in the knowledge of the disease. This suggests the need for more awareness to increase public knowledge and reduce the worries of the Bangladeshi people regarding the SARS-Cov-2 virus. In addition to adhering to the government recommendations of routine hand washing and home quarantine, older males of the Muslim faith could be targeted to further improve the knowledge and avoid further transmission of this novel coronavirus, even as the lockdown continues. The current study provided the first evidence of the knowledge and perception of people using an appropriately sampled population during a critical period—the early stage of the COVID-19 outbreak. However, the online nature of data collection meant that respondents who had an internet connection were more likely to participate, which may lead to bias, including selection bias because of the over-representation of well-educated people in Bangladesh compared to the background population [36] and, as such, the findings may not represent the opinion of the less educated population. Hence, findings from this study cannot be generalizable to the entire Bangladeshi population and lack causal inference because it was an online cross-sectional design. Despite this limitation, this was the only feasible way of data collection at the time of this study. Additionally, since the virus is novel and already widespread, there is little possibility to undertake extensive social studies in Bangladesh. Another limitation of this study was the cross-sectional study design, making it impossible to determine causation. Further studies across randomly selected populations across the country are needed to confirm these findings. Such studies should also assess the social aspects of the condition. Despite these limitations, the present study provides relevant information to fill research gaps in the fight for COVID-19.

## 6. Data Sharing

The datasets analyzed during this study are available from the authors on reasonable request.

## Figures and Tables

**Figure 1 ijerph-17-05252-f001:**
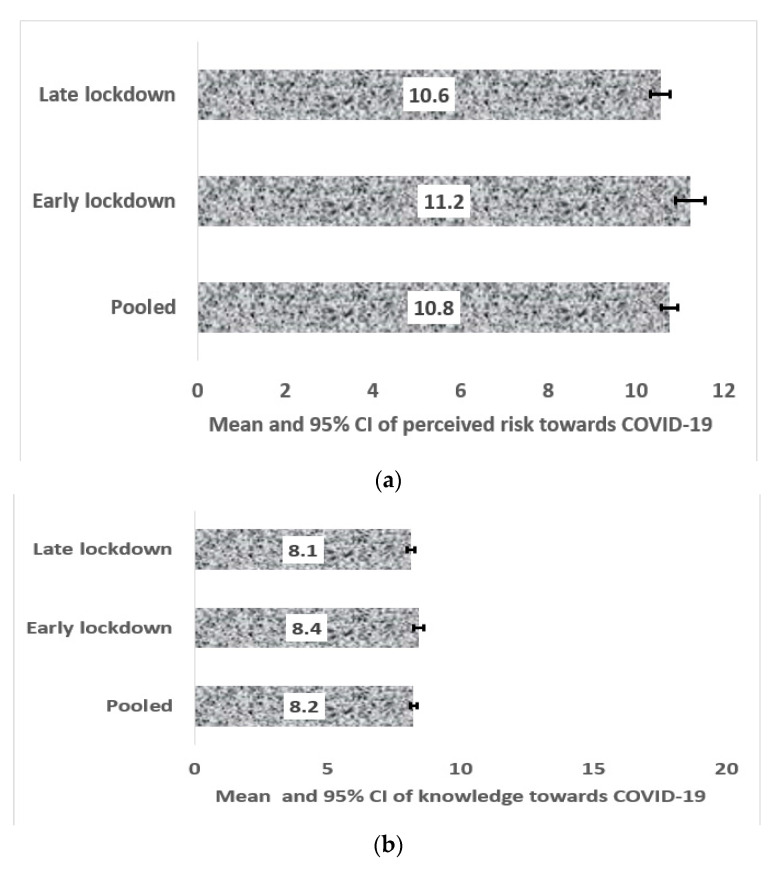
(**a**) Mean and 95% CI of perceived risk towards SARS-Cov-2 in Bangladesh; (**b**) Mean and 95% CI of knowledge towards SARS-Cov-2 in Bangladesh.

**Table 1 ijerph-17-05252-t001:** Questionnaire of knowledge and perception towards COVID-19.

**Knowledge**	
**K1**	Are you aware of the Coronavirus disease (COVID-19) outbreak?
**K2**	Do you think Coronavirus disease (COVID-19) outbreak is dangerous?
**K3**	Do you think Public Health Authorities in Bangladesh are doing enough to control the Coronavirus disease (COVID-19) outbreak?
**K4**	Do you think Hand Hygiene / Hand cleaning is important to control the spread of the Coronavirus disease (COVID-19) outbreak?
**K5**	Do you think wearing masks is important to control the spread of the Coronavirus disease (COVID-19) outbreak?
**K6**	Which mask(s) do you think is better to control the spread of the Coronavirus?
**K7**	Do you think antibiotics can be effective in preventing Coronavirus disease (COVID-19) outbreak?
**K8**	Do you think there are any specific medicines to treat Coronavirus disease (COVID-19)?
**K9**	Those that have contact with someone who has COVID-19 infection should be isolated in the right place immediately. The observation period is usually 14 days
**K10**	Children and young adults should not take steps to prevent the COVID-19 virus from infection.
**K11**	COVID-19 individuals with no symptoms of fever cannot spread the virus to anyone
**K12**	Individuals should stop being crowded to prevent COVID-19 infection.
**Perception**	
Please rate your chances of personal risk of infection with COVID-19 for each of the following?	
**P1**	Risk of becoming infected.
**P2**	Risk of becoming severely infected
**P3**	Risk of dying from the infection
**P4**	How much worried are you because of COVID-19?
**P5**	Are you currently or have you been in (domestic/home) quarantine because of COVID- 19?
**How do you feel about the quarantine?**	
**P6**	I am worried/anxious/alarmed and frightened by the quarantine.
**P7**	I consider the quarantine as necessary and reasonable.
**P8**	I am nervous about the quarantine.
**P9**	I am bored by the quarantine.
**P10**	I am frustrated by the quarantine.
**P11**	I am angry because of quarantine.

Revised and Adopted from World Health Organization, 2019: available at https://www.who.int/bulletin/online_first/20-256651.pdf).

**Table 2 ijerph-17-05252-t002:** Sociodemographic characteristics of the study.

Characteristics	Early Lockdown, *n* (%)	Late Lockdown, *n* (%)	Total, *n* (%)
Responses	322 (32.0)	683 (68.0)	1005 (100.0)
**Demography**			
**Gender**			
Male	163 (50.6)	352 (51.6)	515 (51.3)
Female	159 (49.4)	330 (48.4)	489 (48.7)
**Age stratification (years)**			
18–28	191 (59.3)	341 (50.2)	532 (53.2)
29–38	52 (16.1)	139 (20.5)	191 (19.1)
39–48	53 (16.5)	117 (17.2)	170 (17.0)
49+years	26 (8.1)	82 (12.1)	108 (10.8)
**Education Level**			
Master’s Degree or Equivalent	125 (38.8)	237 (34.7)	362 (36.0)
Bachelor’s Degree	152 (47.2)	321 (47.0)	473 (47.1)
Primary/Secondary	45 (14.0)	125 (18.3)	170 (16.9)
**Marital Status**			
Not married	150 (46.6)	270 (39.5)	420 (41.8)
Married	172 (53.4)	413 (60.5)	585 (58.2)
**Employment Status**			
Unemployed	158 (49.1)	260 (38.1)	418 (41.6)
Employed	164 (50.9)	423 (61.9)	587 (58.4)
**Religious status**			
Muslim	284 (88.2)	608 (89.2)	892 (88.8)
Others (Christian/Hindu)	38 (11.8)	74 (10.9)	112 (11.2)
**Perceived risk of COVID-19**			
**Practice on quarantine**			
Low Practice Quarantine	132 (41.0)	242 (35.4)	374(37.2)
High Practice Quarantine	190 (59.0)	441 (64.6)	631(62.8)
**Current, previously Quarantined for COVID-19**			
Yes, voluntarily	185 (57.5)	473 (69.4)	658 (65.5)
Yes, public health officers request	61 (18.9)	92 (13.5)	153 (15.4)
No	76 (23.6)	117 (17.2)	193 (19.2)
**COVID-19 worries**			
Somehow worried ^$^	223 (69.3)	92 (13.5)	315 (31.3)
Very worried	99 (30.8)	591 (86.5)	690 (68.7)

$ = low, neutral and moderate.

**Table 3 ijerph-17-05252-t003:** Factors associated with the perceived risk of COVID-19 among the respondents in Bangladesh during the lockdown period. (Bold indicates a significant association).

Demography	Coefficient	95 %CI	Adjusted coefficient	95%CI
**Time (Ref = early lockdown)**	**Ref**		**Ref**	
Late lockdown	−0.69	−1.09, −0.29	−0.85	**−1.31, −0.39**
**Gender**				
Male	Ref		Ref	
Female	−0.79	−1.16, −0.42	−0.60	**−0.96, −0.24**
**Age stratification (years)**				
18–28	Ref		-	-
29–38	−0.47	−0.97, −0.03		
39–48	−0.34	−0.87, 0.18		
49+years	−0.78	−1.41, −0.15		
**Education level**				
Master’s Degree or Equivalent	Ref		- -	
Bachelor’s Degree	−0.11	−0.53, 0.31	- -	
Primary/Secondary	−0.19	−0.74, 0.37	- -	
**Marital Status**				
Not married	Ref		- -	
Married	−0.3	−0.68, 0.09	- -	
**Employment Status**				
Unemployed	Ref		- -	
Employed	−0.17	−0.55, 0.21	- -	
**Religious status**				
Muslim	Ref		- -	
Others (Christian/Hindu)	−0.6	−1.19, −0.00	- -	
**Perceived risk of COVID-19**				
**Practice on quarantine**				
Low Practice quarantine	Ref		Ref	
High Practice quarantine	1.18	0.80, 1.57	1.14	0.77, 1.50
**Current, previously Quarantined for COVID-19**				
Yes, voluntarily	Ref		Ref	
Yes, public health officers request	1.57	1.06, 2.09	1.47	**0.97, 1.97**
No	1.85	1.39, 2.32	1.59	**1.17, 3.92**
**COVID-19 worries**				
Somehow worried ^$^	Ref		Ref	
Very worried	0.07	−0.34, 0.47	0.50	**0.04, 0.96**

Note $ = low, neutral, and moderate; CI including “0” indicates non-statistically significant.

**Table 4 ijerph-17-05252-t004:** Factors associated with the knowledge level of COVID-19 among the respondents in Bangladesh during the lockdown period. (Bold indicates a significant association.)

Demography	Coefficient	95 %CI	Adjusted Coefficient	95%CI
Time (Ref = early lockdown)	Ref		Ref	
Late lockdown	−0.29	−0.53, −0.04	−0.22	−0.46, 0.03
Gender				
Male	Ref		Ref	
Female	−0.01	−0.24, 0.22		
**Age stratification (years)**				
18–28	Ref		Ref	
29–38	0.06	−0.25, 0.36	-	-
39–48	0.38	0.06, 0.70	-	-
49+years	0.23	−0.15, 0.61	-	-
**Education level**				
Master’s Degree or Equivalent	Ref		Ref	
Bachelor’s Degree	−0.3	−0.56, −0.05	-0.26	**−0.51, −0.01**
Primary/Secondary	−0.61	−0.95, −0.27	-0.50	**−0.83, −0.16**
**Marital Status**				
Not married	Ref		Ref	
Married	0.05	−0.19, 0.28	-	-
**Employment Status**				
Unemployed	Ref		Ref	
Employed	0.24	0.01, 0.47	-	-
**Religious status**				
Muslim	Ref		Ref	
Others (Christian/Hindu)	0.45	0.09,0.82	0.38	**0.02, 0.74**
**Perceived risk of COVID-19**				
**Practice on quarantine**				
Low Practice quarantine	Ref		Ref	-
High Practice quarantine	−0.16	−0.40, 0.08	1.14	**0.77, 1.50**
**Current, previously Quarantined for COVID-19**				
**Yes, voluntarily**	Ref		Ref	
Yes, public health officers request	0.49	0.17, 0.81	0.42	**0.10, 0.75**
No	0.19	−0.10, 0.48	0.15	−0.14, 0.45
**COVID-19 worries**				
Somehow worried ^$^	Ref		-	
Very worried	−0.16	−0.41, 0.09	-	

$ = low, neutral and moderate.

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
