# Peer review of "Factors Associated with the Perception of Risk and Knowledge of Contracting the SARS-Cov-2 among Adults in Bangladesh: Analysis of Online Surveys"

_ijerph, 2020, doi:10.3390/ijerph17145252_

Round 1

Reviewer 1 Report

I think the manuscript has improved from the first submission. The authors made an effor on replying to my comments. I have some minor changes to suggests: 1) In the abstract the first sentence is wrong. COVID-19 doesn't spread only via human touch. 2) Please avoid indicate COVID-19 as a virus. In some sentences COVID-19 is still indicated as a virus. 3) In the introduction figures are reported to show the burden of the disease. Tehse figures should refer to dates as they change quite quickly. 4) I still don't understand how the sample size paragraph is formulated. Was the sample size calculate in order to reach a given precision of a prevalence of to detect difference on a continuous outcome? 5) It looks that the last section of the questionnaire (How do you feel about quarantine was summarised using a categorical variable. Is it the last variable reported in table 2 (COVID-19 worries). The categories doesn't correspond to those reported in the Independent variables paragraph. The distribution of this variable in the early and late lockdown period is very different. Did the authors used perido specific cut-offs or overall cut-offs? 6) I can't see Figure 1 7) As a general comment the difference detected looks of a small magnitude. Teh authors should be aware that statistical signficance doen't always reflect into meaningfull difference from a clinicla or public health point of view.

Author Response

Reviewer 1

Comments and responses

  1. In the abstract the first sentence is wrong. COVID-19 doesn't spread only via human touch.

Response: We have changed it.

  1. Please avoid indicate COVID-19 as a virus. In some sentences COVID-19 is still indicated as a virus.

Response:  We have changed it.

  1. In the introduction figures are reported to show the burden of the disease. These figures should refer to dates as they change quite quickly.

Response:  We have changed it.

  1. I still don't understand how the sample size paragraph is formulated. Was the sample size calculate in order to reach a given precision of a prevalence of to detect difference on a continuous outcome?

Response: Although the outcome variables are continuous, but the key driver of this research was contracting COVID-19 and policy implication was also geared around reducing contracting COVID-19 virus. Additionally, we added the text below in the manuscript:  “…the key driver of this research was contracting COVID-19 and…”

  1. It looks that the last section of the questionnaire (How do you feel about quarantine was summarized using a categorical variable. Is it the last variable reported in table 2 (COVID-19 worries)? The categories don’t correspond to those reported in the Independent variables paragraph. The distribution of this variable in the early and late lockdown period is very different. Did the authors used period specific cut-offs or overall cut-offs?

Response: we added footnote for clarity.

  1. As a general comment the difference detected looks of a small magnitude. The authors should be aware that statistical significance doesn’t always reflect into meaningful difference from a clinical or public health point of view.

Response: The authors agree with the reviewer’s comments. However, the study assessed the factors associated rather than the differences, indicating the likelihood of an event occurrence.

Reviewer 2 Report

This paper explored the risk perception and knowledge towards COVID-19 among Bangladesh adult participants by two self-administered online surveys administered at two different time of lockdown.

GENERAL COMMENTS

The authors made a mistake about SARS-CoV-2 previously known as 2019-nCoV that can trigger COVID-19. The clinical presentation of SARS-CoV-2 have a very varied range from cases that are asymptomatic or will have mild disease to serious outcomes, which may associated with ARDS. For those symptomatic, fever and cough are the dominant symptoms, as an acute respiratory infectious disease. The severe illness is most likely to develop amongst people of advanced age and those with comorbidities, such are hypertension, diabetes, cardiovascular disease, chronic lung disease, and obesity. To delay the spread of infection and dispersing pressures on hospitals were adopted several public health measures, namely surveillance, contact tracing, social distancing, travel restrictions, and educating the public on hand hygiene. In this paper it was not provided a clear statement of the problem on containment strategies designed to prevent community transmission, and on the patterns of SARS-CoV-2 clinical characteristics.

The authors mentioned that the WHO guidelines for maintaining essential health services during the COVID-19 pandemic were useful for preparing the COVID-19 knowledge questionnaire. The question is, how useful are those WHO guidelines for preparing the questionnaire?

The description of the results of Table 2 in the text is not clear – related with early lockdown, late lockdown or total? Fig 1a and 1b are very difficult to read – what mean the numbers? How these numbers were reached?

The study is not representative of the entire population because the over-representation of well-educated people. So the results obtained in this and in past studies on the perception of risk and knowledge of contracting SARS-CoV-2 are not comparable with those of other countries.

Abstract

Lines 1 and 2 – Coronavirus disease 2019 (COVID-19) is an infectious disease… COVID19 pandemic…

Introduction

1st paragraph

Line 2 – The 2019-nCoV was further named SARS-CoV-2.

The reference should be numbered in order of appearance in the text, so the reference 24 should change to 1 and so on.

Line 3 – SARS-CoV-2 is a betacoronavirus genetic closely linked to the SARS-CoV-1 (79% sequence identity) than to the MERS-CoV (51,8% identity)

Line 5 – The most common symptoms of COVID-19 are fever and cough, and later in the disease, patients are more likely to have difficulty breathing and develop pneumonia. At diagnosis, 80% of cases are asymptomatic or will have mild disease, 15% are severe and 5% are considered critical.

Lines 6 and 7 – There are currently no approved drugs, immune therapies or vaccines against SARS-CoV-2.

2nd paragraph

Line 3 to 5 – Severe illness is most likely to develop amongst people of advanced age and those with co-existing conditions including hypertension, diabetes cardiovascular disease, chronic lung disease, and obesity. The overall mortality rate for COVID-19 will likely be in the range of 0.5% to 3.5%.

Line 7 – …can perish as a result of overwhelm health services.

3rd paragraph

In order to delay the spread of infection and dispersing pressures on hospitals, public health authorities of almost countries adopted vigorous public health measures, such as surveillance, exhaustive contact tracing, social distancing, travel restrictions, and educating the public on hand hygiene, and postponing non-essential operations and services.

Methodology 3.2. Questionnaire

1st paragraph

Lines 3 to 6 – Reference 22 – COVID-19: operational guidance for maintaining essential health services during an outbreak: interim guidance, 25 March 2020. These guidelines provide guidance on a set of targeted immediate actions to reorganize and maintain access to essential quality health services. How these guidelines were useful for the COVID-19 knowledge questionnaire?

Table 1. The source for the questionnaire is correct? – Please detail.

Results 5.1. Descriptive statistics

Table 2 – Age stratification years 18-28 – the number in the text do not match with the number that shows in the table Marital status education level, and practice on quarantine – the numbers and percentages in the text represent only the early lockdown, not the total.

Discussion

1st paragraph

Line 2 and following – There is no Novel COVID-19, but 2019-nCoV or SARS-CoV-2.

2nd paragraph

Line 4 – “…infected individuals and deaths…” instead “…infections individuals and deaths…”.

3rd paragraph

Line 5 and following – “Where according to a study conducted by (20)…”?

“…the risk of contracting the infection caused by COVID-19”. COVID-19 is the disease generated by SARS-CoV-2.

4th paragraph

Line 11 – “…early stage of the virus.” What this means? May be “…the early stage of the pandemic in Bangladesh”.

9th paragraph

Line 1 – “…medical remedies…” may be “…pharmacologic therapies…”.

Author Response

Reviewer 2

This paper explored the risk perception and knowledge towards COVID-19 among Bangladesh adult participants by two self-administered online surveys administered at two different time of lockdown. 

GENERAL COMMENTS

  1. The authors made a mistake about SARS-CoV-2 previously known as 2019-nCoV that can trigger COVID-19. The clinical presentation of SARS-CoV-2 have a very varied range from cases that are asymptomatic or will have mild disease to serious outcomes, which may associate with ARDS. For those symptomatic, fever and cough are the dominant symptoms, as an acute respiratory infectious disease. The severe illness is most likely to develop amongst people of advanced age and those with comorbidities, such are hypertension, diabetes, cardiovascular disease, chronic lung disease, and obesity. To delay the spread of infection and dispersing pressures on hospitals were adopted several public health measures, namely surveillance, contact tracing, social distancing, travel restrictions, and educating the public on hand hygiene. In this paper it was not provided a clear statement of the problem on containment strategies designed to prevent community transmission, and on the patterns of SARS-CoV-2 clinical characteristics.

Response: We have changed it.

  1. The authors mentioned that the WHO guidelines for maintaining essential health services during the COVID-19 pandemic were useful for preparing the COVID-19 knowledge questionnaire. The question is, how useful are those WHO guidelines for preparing the questionnaire?

Response: The survey tool was adapted from the WHO guidelines with little modification to suit the aim of the project in our population.

  1. The description of the results of Table 2 in the text is not clear – related with early lockdown, late lockdown or total?

Response: We have revised the description of Table 2 for clarity. It now reads:

Table 2. Descriptive statistics of the sociodemographic characteristics of the study sample and the summary of responses obtained from those who participated in the survey during the early lockdown (26-31 March 2020), late lockdown (11-16 May 2020) periods. Total responses were a combination of both.

  1. Fig 1a and 1b are very difficult to read – what mean the numbers? How these numbers were reached ?

Response: We have made Figure 1 more legible to read. The mean numbers were the mean scores and how this was calculated have been described in the methods. We have also provided footnote with the figure indicating the minimum and maximum scores of perceived risk and knowledge.

  1. The study is not representative of the entire population because the over-representation of well-educated people. So the results obtained in this and in past studies on the perception of risk and knowledge of contracting SARS-CoV-2 are not comparable with those of other countries.

Response: We have highlighted some of the limitations of the study and this did not include the lack of comparability with other studies from elsewhere. Our study suffers from setbacks faced by most online surveys including the subjectivity of responses obtained via self-administered surveys and the over representation of a cohort of people. While our study can be compared with any study from other countries using similar data collection, the results should be interpreted with caution and in the context of the period of data collection.

In addition, we have revised a section in the limitations for clarity. It now reads:

“However, the online nature of data collection meant that respondents who had internet connection were more likely to participate. This may have led to the over-representation of well-educated people, in Bangladesh compared to the background population [33] and the findings may not represent the opinion of the less educated persons”

Abstract

  1. Lines 1 and 2 – Coronavirus disease 2019 (COVID-19) is an infectious disease… COVID19 pandemic…

Response: We have changed it.

Top of Form

Introduction

  1. 1st Line 2 – The 2019-nCoV was further named SARS-CoV-2.

Response: We have changed it.

  1. The reference should be numbered in order of appearance in the text, so the reference 24 should change to 1 and so on.

Response: we followed the journal’s referencing style.

  1. Line 3 – SARS-CoV-2 is a beta coronavirus genetic closely linked to the SARS-CoV-1 (79% sequence identity) than to the MERS-CoV (51,8% identity)

Response: We have changed it.

  1. Line 5 – The most common symptoms of COVID-19 are fever and cough, and later in the disease, patients are more likely to have difficulty breathing and develop pneumonia. At diagnosis, 80% of cases are asymptomatic or will have mild disease, 15% are severe and 5% are considered critical.

Response: We have changed it.

  1. Lines 6 and 7 – There are currently no approved drugs, immune therapies or vaccines against SARS-CoV-2.

Response: We have changed it. 

  1. 2nd Line 3 to 5 – Severe illness is most likely to develop amongst people of advanced age and those with co-existing conditions including hypertension, diabetes cardiovascular disease, chronic lung disease, and obesity. The overall mortality rate for COVID-19 will likely be in the range of 0.5% to 3.5%.

Response: We have changed it.

  1. Line 7 – …can perish as a result of overwhelm health services.

Response: We have changed it.

  1. 3rd In order to delay the spread of infection and dispersing pressures on hospitals, public health authorities of almost countries adopted vigorous public health measures, such as surveillance, exhaustive contact tracing, social distancing, travel restrictions, and educating the public on hand hygiene, and postponing non-essential operations and services.

 Response: We have changed it.

  1. Methodology2. Questionnaire. 1stparagraph. Lines 3 to 6 – Reference 22 – COVID-19: operational guidance for maintaining essential health services during an outbreak: interim guidance, 25 March 2020. These guidelines provide guidance on a set of targeted immediate actions to reorganize and maintain access to essential quality health services. How these guidelines were useful for the COVID-19 knowledge questionnaire?

Response: Provided in the main manuscript

  1. Table 1. The source for the questionnaire is correct? – Please detail.

Response:  Source has been given in the main manuscript

  1. Results1. Descriptive statistics. Table 2 – Age stratification years 18-28 – the number in the text do not match with the number that shows in the table Marital status education level, and practice on quarantine – the numbers and percentages in the text represent only the early lockdown, not the total.

Response: Thanks, and we have edited the manuscript.

  1. 1stparagraph. Line 2 and following – There is no Novel COVID-19, but 2019-nCoV or SARS-CoV-2.

Response: We have changed it.

  1. 2nd Line 4 – “…infected individuals and deaths…” instead “…infections individuals and deaths…”.

Response: We have changed it.

  1. 3rd Line 5 and following – “Where according to a study conducted by (20)…”? “…the risk of contracting the infection caused by COVID-19”. COVID-19 is the disease generated by SARS-CoV-2.

Response: We have changed it.

  1. 4th Line 11 – “…early stage of the virus.” What this means? May be “…the early stage of the pandemic in Bangladesh”.

Response: We have changed it.

  1. 9th Line 1 – “…medical remedies…” may be “…pharmacologic therapies…”.

Response: We have changed it.

Reviewer 3 Report

Overall this is good information and should be shared with strong caveats. However, several items should be addressed before publication is considered. The overarching question is, what can this data tell us based on the sampling frame? There is a strong selection bias, meaning these results may not reflect the adult population of Bangladesh. Moreover, these results may confound or highlight perceptions that are not true of the population, leaving readers of this article to believe the results say one thing when they in actuality say the opposite. This is always the challenge of selection bias.

Abstract:

  • Very difficult to follow in sections and to understand what the authors are trying to convey with their results.

Introduction:

  • Citations are out of order.
  • The number from the pandemic is already outdated as I read these, so it would be good to inform the reader of the date these numbers were reported to provide the context in this timeframe.
  • Highlighting in odd places in the document?

Methods:

  • Different font and sizes throughout.
  • Further explanation is needed for why splitting the worried groups arbitrarily into three categories.
  • There are always issues epidemiologically by letting the model inform which variables to include in the multivariable regression model. Much better to use information from other sources and what is known about these other factors then decide which variables to include or not. More justification is needed to keep this as is.

Results:

  • Figures 1a and 1b are difficult to understand. Suggest revising.

Discussion:

  • On page 9, “Where according to a study conducted by [20]…” this needs to say the author's name. In the following sentences, the word “huge” is used, suggest replacing.
  • Good discussion in general, but needs more focused attention to the main take-home messages…what are the results that can have action taken?

Author Response

Reviewer 3

  1. Overall this is good information and should be shared with strong caveats. However, several items should be addressed before publication is considered. The overarching question is, what can this data tell us based on the sampling frame? There is a strong selection bias, meaning these results may not reflect the adult population of Bangladesh. Moreover, these results may confound or highlight perceptions that are not true of the population, leaving readers of this article to believe the results say one thing when they in actuality say the opposite. This is always the challenge of selection bias.

Response:  As part of the limitation including selection bias, we added the text below:

  1. However, the online nature of data collection meant that respondents who had internet connection were more likely to participate which may lead to bias including selection bias because of the over-representation of well-educated people, in Bangladesh compared to the background population [33] and as such the findings may not represent the opinion of the less educated persons. Hence, findings from this study cannot be generalisable to the entire Bangladeshi population and lack causal inference because it was an online cross-sectional design. Despite this limitation, this was the only feasible way of data collection at the time of this study.

              Response: We have rationalized it

  1. Abstract: Very difficult to follow in sections and to understand what the authors are trying to convey with their results.

Response:  We have edited the abstract for clarity

  1. Introduction: Citations are out of order.

Response: We followed the referencing style of the journal

  1. The number from the pandemic is already outdated as I read these, so it would be good to inform the reader of the date these numbers were reported to provide the context in this timeframe.

Response: We have changed it.

  1. Highlighting in odd places in the document?

Response: We have changed it.

  1. Methods: Different font and sizes throughout.

Response: We have changed it.

  1. Further explanation is needed for why splitting the worried groups arbitrarily into three categories.

Response:  Our apologies, we don’t understand your question, but the worries group was in two categories and we have added a footnote indicating somehow worried was combined.

  1. There are always issues epidemiologically by letting the model inform which variables to include in the multivariable regression model. Much better to use information from other sources and what is known about these other factors then decide which variables to include or not. More justification is needed to keep this as is.

Response: We have added the text below in the manuscript. In our regression analysis, we checked for homogeneity of variance and multicollinearity using Variance Inflation Factors (VIF).

  1. Results: Figures 1a and 1b are difficult to understand. Suggest revising.

Response: Done. See response above

  1. Discussion: On page 9, “Where according to a study conducted by [20]…” this needs to say the author's name. In the following sentences, the word “huge” is used, suggest replacing.

Response: We have changed it.

  1. Good discussion in general, but needs more focused attention to the main take-home messages…what are the results that can have action taken?

Response: We have changed it.

In addition, in the key text, we updated all evaluation remarked by the respected reviewers and did the requisite correction. All authors in this study have agreed upon its’ revised manuscript version. Later we did some copy editing for more manuscript standardization and by doing so we use all necessary jargons with care.  

Round 2

Reviewer 2 Report

The manuscript benefit of the review done by the authors, and was substantially improved.

The abbreviation 2019-nCoV should switch to SARS-Cov-2 in order to standardize the abbreviated name of the virus.

  1. Methodology
Line 109 – “late lockdown”

Line 117 – (early lockdown)

Line 121 – (late lockdown)

  1. Results
Line 180 – lockdown periods

Line 190 – Figure 1a and 1b

Table 2 – Age stratification (18-28 years) percentages are (and text) not correct

  1. Discussion
Line 258 – (SARS-CoV-1)

Line 301 – SARS-CoV-1 in 2003

Author Response

Reviewer 2

The manuscript benefit of the review done by the authors, and was substantially improved.
Response: Thank you for the compliment.

The abbreviation 2019-nCoV should switch to SARS-Cov-2 in order to standardize the abbreviated name of the virus.

Response: Revised in the attachment manuscript.

  1. Methodology

Line 109 – “late lockdown”

Response: Revised in the attachment manuscript.

Line 117 – (early lockdown)
Response: Revised in the attachment manuscript.

Line 121 – (late lockdown)
Response: Revised in the attachment manuscript.

  1. Results

Line 180 – lockdown periods
Response: Revised in the attachment manuscript.

Line 190 – Figure 1a and 1b
Response: Revised in the attachment manuscript.

Table 2 – Age stratification (18-28 years) percentages are (and text) not correct
Response Revised in the attachment manuscript.

  1. Discussion

Line 258 – (SARS-CoV-1)
Response: Revised in the attachment manuscript.

Line 301 – SARS-CoV-1 in 2003

Response: Revised in the attachment manuscript.

Reviewer 3 Report

The changes and edits address all of the comments but for one. 

  1. There are always issues epidemiologically by letting the model inform which variables to include in the multivariable regression model. Much better to use information from other sources and what is known about these other factors then decide which variables to include or not. More justification is needed to keep this as is.

Response: We have added the text below in the manuscript. In our regression analysis, we checked for homogeneity of variance and multicollinearity using Variance Inflation Factors (VIF).

Please provide more rationale for WHY you included your variables in the model.

Author Response

Reviewer 3

The changes and edits address all of the comments but for one. 

Question: Please provide more rationale for WHY you included your variables in the model.

Response: Research on COVID-19 are still very limited including no Causal research (cause-and-effect relationships) has been conducted in Bangladesh and research on COVID-19 are mostly online research which may be biased towards the computer literates in the country. Including variable based on previous research maybe too premature because Research on COVID-19 are still new and subject to further investigation.

Question: There are always issues epidemiologically by letting the model inform which variables to include in the multivariable regression model. Much better to use information from other sources and what is known about these other factors then decide which variables to include or not. More justification is needed to keep this as is.

Response: We agreed with the reviewer and we have conducted more statistical analysis to justify the variables we have as our final model and in addition, the text below has been added to both the statistical and result section of the manuscript.

Additionally, and to double check our linear regression output, we performed a similar stage modelling employed by Dibley et al. [2012]. A 2-staged modelling technique was employed in the multivariable modelling. In the first stage, the demography factors were entered into the baseline multivariable model. A manual process backward elimination was performed, and variables with p <0.05 were retained in the first model (Model 1). In the second and final stage modelling, perceived risk of COVID-19 factors were added into significant variables in Model 1, and variables with p-values < 0.05 were retained in the final model.

Dibley, M.J.; Titaley, C.R.; d’Este, C.; Agho, K. Iron and folic acid supplements in pregnancy improve child survival in Indonesia. Am. J. Clin. Nutr. 201295, 220–230

Age stratification was significant in the univariate analysis and in the final model, we removed religion and replaced with age stratification and the result showed that age stratification was not statistically significant (Wald χ2 = 0.46, P=0.7137) and similarly, when gender and replaced with age stratification but age stratification was not significant (Wald χ2 = 0.49, P=0.6908). 

Age stratification and employment status were significant in the univariate analysis and in our final model, we removed religion and replaced it with age stratification and age stratification was not statistically significant (Wald χ2 = 1.44, P=0.2293) and when education was replaced with age stratification, age stratification was not significant (Wald χ2 = 2.54, P=0.055) but when education was replaced by employment status, employment status was associated with increased knowledge of COVID-19 [β=0.26, 95%CI: 0.03, 0.49, P=0.027 for those employed}

This manuscript is a resubmission of an earlier submission. The following is a list of the peer review reports and author responses from that submission.

Round 1

Reviewer 1 Report

Despite of the relevance of the topic and the appropriate design of epidemiological study chosen, the study has a very small scientific value. The main issue is related to inappropriate selection of the study group and very small sample size of the country that has around 165 million of people. The study group does not represent the country's population neither by gender, nor by age. Most of the subjects were highly educated people (limitation mentioned). Therefore, it is too strong to say that evaluation of the perception of risk for contracting COVID-19, and the awareness level of people in Bangladesh regarding the infection was made. 

Multivariate logistic regression used for the analyses described and presented improperly. Therefore, it is not understandable what model was used to calculate ORa.

The conclusion made could have been made without any research.

Reviewer 2 Report

Major remarks

Major remarks

Methods

Selection of sample is not clear.  Although authors make it clear that it was a convenience sample of 385 respondents of Bangladesh nationality and residing in Bangladesh   at the time of this study were selected,  several further details provided  are not informative.  What means that “Respondents had selected such that there was approximately  the same distribution by age, sex, and ethnicity”

The next sentence is also not clear  - “. Specifically, a population stratum was established with a predetermined number of open slots into which eligible respondents in the online pool could  enrol on a first-come, first-serve basis. “

What means that “ This study assumed a proportion of 50% of the population and then based the calculation of total sample size on desire precision of 5% and 5% significance level for a two-sided test?

“What is “Likert scale with five levels”?

The worried about quarantine score – means score indicating being worried about quarantine?

It is not clear how the results were interpreted. How criterion for adequate knowledge and adequate perception were created.

Authors provide following description :” The explanatory (independent) variable included “high quarantine practice” which was derived  by combining the average quarantine practice (33.3%) with the high quarantine practice (33.3%), and  low quarantine practice was “low quarantine practice scores” 33.3%.”

 Are the percentages (33.3%) referring to responders ?

Minor remarks:

Results

I would suggest following changes in description of results

Respondents who went into quarantine as directed by the public health team, and those who did not  quarantine at all, had increased odds of the high perceived risk of contracting COVID- 19 compared to  those that voluntarily quarantined themselves during the study period (aOR 2.85, 95%CI 1.47-5.53) and those that did not quarantine themselves (aOR 2.14, 95%CI 1.17-3.92) repectively

Discussion

Some parts of discussion should be rewritten as they are difficult to read.

Example: “While the present research showed a lower score of knowledge among Bangladesh citizens, it was surprising to find average score of knowledge against  COVID-19, as this epidemiological survey was performed at the very early stage of the virus. We  believe this to be partially attributed to the survey being skewed by people with a Bachelor degree or higher, the large percentage of respondents being 86 per cent. “

The other sentence which is difficult to read.

The slightly higher  chances of sufficient information among citizens who did not quarantine themselves relative to those  who did so willingly can be due to the less severe situation of the COVID-19 outbreak in Bangladesh  and the prevalence of younger adults in this sample, resulting in respondents feeling that they have a  lower probability of contamination with the COVID-19 virus.

Conclusions

In my opinion ,  authors  conclusion referring to the fact that “ higher COVID-19 awareness scores are strongly correlated with not becoming a religious believer” is very important and have several implications for controlling COVID-19 epidemic in Bangladesh

Reviewer 3 Report

This study present the results on a survey conducted to measure knowledge and risk perception on COVID-19 and their association with socio-economic variables. The survey also evaluate feelings about quarantine measures and their association knowledge and risk perception on COVID-19. The survey was conducted in Bangladesh during the lockdown period.

I think the research question is interesting as could help public health officer to focus messages on particular groups, but the research and the draft have several weakness that need to be solved before the paper could be submitted to this and other journals.

I think there are two main issues in the research method:

  1. The survey was conducted via web and posted through social media (facebook, whatsapp). This procedure produce a sample that could not be representative. The authors stratified by age, sex and ethnicity trying to mitigate the selection bias, but it seems that the sample is not representative for educational level. This could reduce the external validity of the results presented in the paper.
  2. The authors derived knowledge and perception scales in a quantitative scale and then categorised them using percentiles. My understanding is that tertiles (33% and 66%) have been used for perception scales and quartiles (25%, 50%, 75%) for the knowledge scale. The interpretation of the categories depend on the distribution of the original scale, that is, if the distribution is approximately normal and symmetric, or left or right skewed. Names assigned to categories are arbitrary looks arbitrary and cannot correspond to what originally measured. That is if the risk perception scale is symmetric with high central value, some of the subject included in the low-risk category could have a high score. More importantly, using percentiles to define the categorical outcome imply that prevalence of the outcome is determined by the authors. That is the 62% adequate risk perception prevalence is due to the fact that the last two textiles were combined.

The manuscript has several issues that need to be addressed:

Introduction

  • Line 52: COVID-19 is the disease not the virus, the virus is SARS-COV-2
  • Line 53: Actually the mortality rate is higher for MARS and SARS
  • Lines 70-80: here there a lot of figures for a set of countries. I would just put the context of COVID-19 outbreak in Bangladesh
  • Lines 83-85: I found the sentence out of the context

Sample size

  • Line 114-117: it is not clear if the sample size was calculated to have a given precision (5%) for the point estimate of the prevalence (assumed 50%) or to compare prevalence rates (what was the minimum difference detectable and the power?).

Methodology

  • Sampling: it seems that the questionnaire has been posted via facebook and whatsapp starting from contacts of the researches involved in this study. Is that the case?
  • Lines 143-145: To categorise the knowledge scale, were used tertiles or quartile? The results presented in figure 1 suggests that quartiles were used.
  • Lines 150-152: There is no need to specify 33% if tertiles were used. I think categories names are wrongly assigned here.
  • Lines 157-160: This scale seems to me related to anxiety or worrying feeling on quarantine measures and not on the adherence to quarantine measures. I would rename the categories. As before, there is no need to specify 33% if tertiles were used.
  • Line 164: I would put “summarised” instead of “presented”
  • Line 166: This is wrong, prevalences of categorical variables category shouldn’t be compared but just described. Overlapping of CI’s is not a inferential procedure.
  • Lines 167-177: no need to repeat 33%
  • Those that are or were previously under quarantine could have/had COVID-19? This could influence their perception of risk and knowledge about COVID-19.
  • Line 192: Figure 1 is very poor. I would remove it. These are prevalence estimates cannot be compared. Cross of the CI is not an inferential procedure
  • Table 3: why there is perceived risk on the rows?

Reviewer 4 Report

The manuscript that I reviewed “Factors associated with adequate perception of risk and knowledge of contracting the Novel COVID-19 among adults in Bangladesh: Analysis of an online cross-sectional survey” is a study aimed to explored the actual perception of knowledge, risk and awareness to develop COVID-19 among Bangladeshi participants in order to  implement emergency policies to counter the spread and impact of the disease.

Major comments

The manuscript is well written and supported by the present data. Furthermore, the purpose appears useful and interesting. In my opinion, I would have added as demographic variable to be parents since the perspective of live usually changes.

Furthermore, I think that the Authors, throughout the manuscript, are often inconsistent in the appropriate employ of “COVID-19” for the disease or the virus. To be clear, initially, the virus was called 2019-nCoV and subsequently SARS-CoV-2 while "COVID-19" is the acronym of "COronaVIrus DIsease 2019”. Finally, I think that the list of reference is not corrected since references must be numbered in order of appearance in the text and listed individually at the end of the manuscript.

Minor comments

1)Line 51: I suggest to the Authors to replace “MARS and SARS” with Middle East Respiratory Syndrome (MERS) and Severe Acute Respiratory Syndrome (SARS) coronaviruses (CoVs).

2)Line 51: I suggest to the Authors to delete this sentence “but much deadlier than both [34].” or to support it with much more references.

3)” P5 Are you currently or have you been in (domestic/home) quarantine because of COVID- 19?”, in my opinion this question is not useful for the purpose since is not a chances of personal risk of infection with COVID-19 but the quarantine status could be consider as a demographic variable of the participants?

Dear Editor,

many thanks for the invitation to review a manuscript to publish on "International Journal of Environmental Research and Public Health”.

The manuscript that I reviewed “Factors associated with adequate perception of risk and knowledge of contracting the Novel COVID-19 among adults in Bangladesh: Analysis of an online cross-sectional survey” is a study aimed to explored the actual perception of knowledge, risk and awareness to develop COVID-19 among Bangladeshi participants in order to  implement emergency policies to counter the spread and impact of the disease.

Major comments

The manuscript is well written and supported by the present data. Furthermore, the purpose appears useful and interesting. In my opinion, I would have added as demographic variable to be parents since the perspective of live usually changes.

Furthermore, I think that the Authors, throughout the manuscript, are often inconsistent in the appropriate employ of “COVID-19” for the disease or the virus. To be clear, initially, the virus was called 2019-nCoV and subsequently SARS-CoV-2 while "COVID-19" is the acronym of "COronaVIrus DIsease 2019”. Finally, I think that the list of reference is not corrected since references must be numbered in order of appearance in the text and listed individually at the end of the manuscript.

Minor comments

1)Line 51: I suggest to the Authors to replace “MARS and SARS” with Middle East Respiratory Syndrome (MERS) and Severe Acute Respiratory Syndrome (SARS) coronaviruses (CoVs).

2)Line 51: I suggest to the Authors to delete this sentence “but much deadlier than both [34].” or to support it with much more references.

3)” P5 Are you currently or have you been in (domestic/home) quarantine because of COVID- 19?”, in my opinion this question is not useful for the purpose since is not a chances of personal risk of infection with COVID-19 but the quarantine status could be consider as a demographic variable of the participants?

Best regards,

Federica Di Profio

Faculty of Veterinary Medicine

University of Teramo 64100

Teramo, Italy